# Association between Severity of Diffuse Idiopathic Skeletal Hyperostosis and Ossification of Other Spinal Ligaments in Patients with Ossification of the Posterior Longitudinal Ligament

**DOI:** 10.3390/jcm10204690

**Published:** 2021-10-13

**Authors:** Soraya Nishimura, Takashi Hirai, Narihito Nagoshi, Toshitaka Yoshii, Jun Hashimoto, Kanji Mori, Satoshi Maki, Keiichi Katsumi, Kazuhiro Takeuchi, Shuta Ushio, Takeo Furuya, Kei Watanabe, Norihiro Nishida, Takashi Kaito, Satoshi Kato, Katsuya Nagashima, Masao Koda, Hiroaki Nakashima, Shiro Imagama, Kazuma Murata, Yuji Matsuoka, Kanichiro Wada, Atsushi Kimura, Tetsuro Ohba, Hiroyuki Katoh, Masahiko Watanabe, Yukihiro Matsuyama, Hiroshi Ozawa, Hirotaka Haro, Katsushi Takeshita, Yu Matsukura, Hiroyuki Inose, Masashi Yamazaki, Kota Watanabe, Morio Matsumoto, Masaya Nakamura, Atsushi Okawa, Yoshiharu Kawaguchi

**Affiliations:** 1Department of Orthopedic Surgery, Keio University School of Medicine, Shinjuku-ku, Tokyo 160-8582, Japan; soraya.nishimura@gmail.com (S.N.); kw197251@keio.jp (K.W.); morio@a5.keio.jp (M.M.); masa@a8.keio.jp (M.N.); 2Department of Orthopedic Surgery, Tokyo Medical and Dental University, Bunkyo-ku, Tokyo 113-8510, Japan; hirai.orth@tmd.ac.jp (T.H.); yoshii.orth@tmd.ac.jp (T.Y.); 0123456789jun@gmail.com (J.H.); ushiorth20@gmail.com (S.U.); matsukura.orth@tmd.ac.jp (Y.M.); inose.orth@tmd.ac.jp (H.I.); okawa.orth@tmd.ac.jp (A.O.); 3Department of Orthopaedic Surgery, Shiga University of Medical Science, Ōtsu 520-2192, Japan; kanchi@belle.shiga-med.ac.jp; 4Department of Orthopedic Surgery, Chiba University Graduate School of Medicine, Chiba-shi 260-0856, Japan; satoshi.maki@chiba-u.jp (S.M.); takeo251274@yahoo.co.jp (T.F.); 5Department of Orthopedic Surgery, Niigata University Medical and Dental General Hospital, Niigata-shi 951-8520, Japan; kkatsu_os@yahoo.co.jp (K.K.); keiwatanabe_39jp@live.jp (K.W.); 6Department of Orthopedic Surgery, National Hospital Organization Okayama Medical Center, Okayama-shi 701-1192, Japan; takeuchi@okayamamc.jp; 7Department of Orthopedic Surgery, Yamaguchi University Graduate School of Medicine, Ube 755-8505, Japan; nishida3@yamaguchi-u.ac.jp; 8Department of Orthopaedic Surgery, Osaka University Graduate School of Medicine, Suita 565-0871, Japan; takashikaito@gmail.com; 9Department of Orthopedic Surgery, Graduate School of Medical Sciences, Kanazawa University, Kanazawa 920-8641, Japan; skato323@gmail.com; 10Department of Orthopedic Surgery, Faculty of Medicine, University of Tsukuba, Tsukuba 305-8577, Japan; katsu_n103@yahoo.co.jp (K.N.); masaokod@gmail.com (M.K.); masashiy@md.tsukuba.ac.jp (M.Y.); 11Department of Orthopaedics, Nagoya University Graduate School of Medicine, 65 Tsurumai, Shouwa-ku, Nagoya 464-8601, Japan; hirospine@med.nagoya-u.ac.jp (H.N.); imagama@med.nagoya-u.ac.jp (S.I.); 12Department of Orthopedic Surgery, Tokyo Medical University, Shinjuku, Tokyo 160-8402, Japan; kaz.mur26@gmail.com (K.M.); yuji_kazu77@yahoo.co.jp (Y.M.); 13Department of Orthopedic Surgery, Hirosaki University Graduate School of Medicine, Hirosaki 036-8562, Japan; wadak39@hirosaki-u.ac.jp; 14Department of Orthopedics, Jichi Medical University, Shimotsuke 329-0498, Japan; akimura@jichi.ac.jp (A.K.); dtstake@gmail.com (K.T.); 15Department of Orthopedic Surgery, University of Yamanashi, Chuo 409-3898, Japan; tooba@yamanashi.ac.jp (T.O.); haro@yamanashi.ac.jp (H.H.); 16Department of Orthopedic Surgery, Surgical Science, Tokai University School of Medicine, Isehara 259-1193, Japan; hero@tokai-u.jp (H.K.); masahiko@is.icc.u-tokai.ac.jp (M.W.); 17Department of Orthopedic Surgery, Hamamatsu University School of Medicine, Hamamatsu 431-3125, Japan; spine-yu@hama-med.ac.jp; 18Department of Orthopaedic Surgery, Tohoku Medical and Pharmaceutical University, Sendai 981-8558, Japan; hozawa@med.tohoku.ac.jp; 19Department of Orthopedic Surgery, Faculty of Medicine, University of Toyama, Toyama-shi 930-0194, Japan; zenji@med.u-toyama.ac.jp

**Keywords:** cervical ossification of the posterior longitudinal ligament, diffuse idiopathic skeletal hyperostosis, whole-spine computed tomography, grading system, multicenter study

## Abstract

Background: Although diffuse idiopathic skeletal hyperostosis (DISH) is known to coexist with the ossification of spinal ligaments (OSLs), details of the radiographic relationship remain unclear. Methods: We prospectively collected data of 239 patients with symptomatic cervical ossification of the posterior longitudinal ligament (OPLL) and analyzed the DISH severity on whole-spine computed tomography images, using the following grades: grade 0, no DISH; grade 1, DISH at T3–T10; grade 2, DISH at both T3–T10 and C6–T2 and/or T11–L2; and grade 3, DISH beyond C5 and/or L3. Ossification indices were calculated as the sum of vertebral and intervertebral levels with OSL for each patient. Results: DISH was found in 107 patients (44.8%), 65 (60.7%) of whom had grade 2 DISH. We found significant associations of DISH grade with the indices for cervical OPLL (r = 0.45, *p* < 0.0001), thoracic ossification of the ligamentum flavum (OLF; r = 0.41, *p* < 0.0001) and thoracic ossification of the supra/interspinous ligaments (OSIL; r = 0.53, *p* < 0.0001). DISH grade was also correlated with the index for each OSL in the whole spine (OPLL: r = 0.29, *p* < 0.0001; OLF: r = 0.40, *p* < 0.0001; OSIL: r = 0.50, *p* < 0.0001). Conclusion: The DISH grade correlated with the indices of OSL at each high-prevalence level as well as the whole spine.

## 1. Introduction

Ossification of the posterior longitudinal ligament (OPLL) is a well-known cause of severe myelopathy and radiculopathy, especially in East Asian countries [1,2]. Patients with OPLL often experience the ossification of spinal ligaments (OSLs). Previous reports suggest that the co-morbidity rate for diffuse idiopathic skeletal hyperostosis (DISH) and OPLL is around 25–50%, which is relatively high [3,4,5,6]. Given that DISH is usually found as a benign radiological condition that does not compress the spinal cord [7,8,9,10], this pathology has been considered clinically innocuous. However, patients with DISH are at higher risk for late-onset paralysis following ankylosing spinal fractures with minor trauma, especially in cases with spinal cord compression due to OPLL [11,12,13]. In addition, myelopathy frequently results from a concentration of stress factors—when spinal stenosis along with OSL is present above or below the ankylosing spine in DISH [14]. Therefore, assessing the degree of DISH is important in patients with cervical OPLL.

Despite the potentially devastating consequences of comorbid DISH and an additional OSL, such as cervical OPLL, a correlation remains to be determined between DISH severity and a predisposition to other OSL. To address this question, a tool is urgently needed for evaluating the spread of DISH. Previous studies have reported the degree of DISH according to the number of consecutive vertebral bodies involved, or the width and/or thickness of ossification on plain radiographs [15,16,17]; however, neither of these grading methods can accurately assess the development of ossified lesions.

In a previous study, we retrospectively examined the DISH distribution pattern in whole-spine computed tomography (CT) images for patients with cervical OPLL [6] and found that DISH developed at the thoracic level initially and extended to the cervical and/or lumbar spine over time. Therefore, we developed a novel four-point grading system that can evaluate the age-related progression of DISH (grade 0, DISH anywhere in the spine; grade 1, DISH at T3–T10; grade 2, DISH extending to the cervicothoracic junction (C6–T2) and/or thoracolumbar junction (T11–L2); grade 3, DISH extending to the cervical and/or lumbar spine beyond C5 and/or L3; Figure 1). At the Japanese Multicenter Research Organization for Ossification of the Spinal Ligament (JOSL), we established a nationwide patient registry to prospectively collect clinical and radiological data, including whole-spine CT scans of OPLL patients, with the aim of clarifying associations with the presence of each type of OSL. Accordingly, the aim of the present study was to investigate the relationship between the severity of DISH (the DISH grade [6]) and all other types of OSL based on the data collected in the patient registry.

## 2. Materials and Methods

### 2.1. Patients and Methods

This multicenter prospective observational cross-sectional study was performed by the JOSL with the assistance of the Japanese Ministry of Health, Labour, and Welfare. The inclusion criteria were as follows: over 20 years of age; diagnosis of cervical OPLL on plain radiographs; symptoms such as neck pain, numbness in the upper or lower extremities, clumsiness, or gait disturbance; presentation to 1 of 16 institutions affiliated with the JOSL between September 2015 and December 2017; and whole-spine CT images available. Patients were excluded if they had undergone surgery to treat OPLL. The study included 239 Japanese subjects (163 men and 76 women). Basic clinical data for age, sex, body mass index (BMI), presence or absence of diabetes mellitus (DM), family history (FH) of OPLL, trauma history (TH), patients with or without surgical treatment, surgical methods and perioperative complications were obtained from patient records held at participating institutions. The study was approved by the institutional review board of each participating institution and was conducted in accordance with all relevant guidelines and regulations.

### 2.2. Radiographic Examinations

Six senior spine surgeons (S.U., K.M., S.M., K.K., N.N. and K.T.) independently determined the incidence of OPLL, ossification of the ligamentum flavum (OLF), ossification of the supra/interspinous ligaments (OSIL), ossification of the anterior longitudinal ligament (OALL), and ossification of the nuchal ligament (ONL) in whole-spine mid-sagittal CT images (Figure 2). Before the evaluation, inter-observer agreement was determined by assessing the incidence of OPLL and OALL, using CT images from the same 10 patients. The average kappa (κ) coefficients of inter-observer agreement for OPLL and OALL were 0.83 and 0.78, respectively. The prevalence rate of ONL was calculated for DISH grades 0 to 3, as described below. We recorded the presence of OPLL, OLF and OSIL for all vertebral bodies and intervertebral disc levels of the whole spine. An ossification index was calculated according to the number of levels with OPLL (OPLL index), OLF (OLF index), or OSIL (OSIL index), as described previously [6,18,19,20]. OALL was considered DISH if it completely bridged at least four contiguous adjacent vertebral bodies in the thoracic spine, according to the criteria established by Resnick and Niwayama (Figure 2) [21]. DISH was classified as follows: grade 0, no DISH at any spine level; grade 1, DISH at T3–T10; grade 2, DISH at both T3–T10 and C6–T2 and/or T11–L2; grade 3, DISH extending beyond C5 and/or L3.

### 2.3. Statistical Analysis

All data are presented as the mean ± standard deviation. Correlations between DISH grade and age, BMI, OPLL index, OLF index, and OSIL index were analyzed using the Pearson’s correlation coefficient. The chi-squared test was used to examine differences in the prevalence rate of ONL, sex distribution, the presence of DM, FH of OPLL, TH, the number of patients treated surgically, the rate of each surgical method, and each complication rate. A *p*-value of <0.01 was considered statistically significant.

## 3. Results

### 3.1. Demographic Data and Surgery-Related Data According to DISH Grade

DISH was observed in 82 men and 25 women with cervical OPLL, with a co-morbidity rate of 44.8% (107/239; Table 1). Our grading system evaluation revealed that when DISH was present, grade 2 was the most common (65/107, 60.7%), followed by grade 1 (23/107, 21.5%) and grade 3 (19/107, 17.8%). There was a slight, yet significant, correlation of DISH grade with age (r = 0.30, *p* < 0.0001; Table 1) but not with sex, BMI, or the prevalence rate of DM, FH of OPLL or TH. Only one case was found in which the bridging of OALL over four adjacent vertebral bodies was localized in the cervical spine. This case was, therefore, excluded from the analysis of patients with DISH because it did not exhibit similar bridging in the thoracic spine. 

Surgical treatment was performed in 59.4% of all cases (142/239; Table 1) in at least one of the spinal levels. The cervical spine was the most frequently treated level (129/239, 54.0%), followed by the thoracic (26/239, 10.9%) and lumbar spine (11/239, 4.6%). There was no significant difference in the rate of surgical treatment between each grade at any spinal level. Laminoplasty was the most common surgical procedure performed on the cervical spine (60/129, 46.5%) whereas posterior decompression with fusion (PDF) was more common at the thoracic spine (16/26, 61.5%). On the other hand, laminectomy and PDF were equally common at the lumbar spine (5/11, 45.5%). No remarkable differences were found in the rates of these procedures between each grade. Furthermore, all the incidences of perioperative complication were not statistically different among the grades.

### 3.2. Association between DISH Grade and OSL

Next, we calculated the correlation coefficient between the DISH grade and OSL for each spinal level. At the cervical level, the DISH grade was moderately correlated with the OPLL index (r = 0.45, *p* < 0.0001; Figure 3a); however, there was no correlation between DISH grade and the OLF index (r = 0.14, *p* = 0.03; Figure 3b). Moreover, the prevalence of ONL was significantly associated with DISH grade (*p* = 0.003, chi-squared test; Figure 3c). At the thoracic spine, the DISH grade was moderately correlated with the OLF and OSIL indices (OLF: r = 0.41, *p* < 0.0001, Figure 4b; OSIL: r = 0.53, *p* < 0.0001; Figure 4c), but not with OPLL (r = 0.12, *p* = 0.06; Figure 4a). There was no significant correlation of DISH grade with any OSL at the lumbar spine (OPLL: r = −0.02, *p* = 0.78, OLF: r = 0.11, *p* = 0.11, OSIL: r = 0.14, *p* = 0.03; Figure 5a–c). Finally, there were moderate to weak correlations between DISH grade and OPLL, OLF and OSIL indices in the whole spine (OPLL: r = 0.29, *p* < 0.0001, OLF: r = 0.40, *p* < 0.0001, OSIL: r = 0.50, *p* < 0.0001; Figure 6a–c).

### 3.3. Case Presentation

A 66-year-old man presented to one of our institutions with difficulty walking. Whole-spine CT imaging showed continuous-type cervical OPLL at C3–C7, with a cervical OPLL index of 10. In addition, extensive thoracic OLF was found, with a thoracic OLF index of 9. Grade 3 DISH was distributed from C4 to L2. The level of maximum compression in the spinal canal was C3/4 with OPLL (Figure 7). Therefore, we decided to perform a two-stage surgery for cervical OPLL. First, anterior decompression with fusion (ADF) was performed from C2 to C5 with grafted bone harvested from the fibula. Two weeks after the initial surgery, an additional posterior fixation was performed from C2 to C7. Five years after the surgeries, the man’s neurological symptoms have shown satisfactory improvement.

## 4. Discussion

In a previous study, we reported on the distribution of DISH in patients with cervical OPLL by cluster analysis [6]. In that study, DISH was found to be gradually distributed from the thoracic to the cervical and lumbar spine, and rarely extended beyond C5 and L3 [6]. Based on these findings, we defined DISH found only in the thoracic spine as a mild case, with C5 and L3 indicating the boundaries between moderate and severe cases. The present study found a weak but significant correlation between DISH grade and age. In addition, there was only one case in which bridging of OALL over four or more vertebral bodies was found in the cervical spine but not in the thoracic spine. These findings support the rationale of our grade, that DISH mainly develops from the thoracic spine to the cervical and lumbar spine over time; therefore, our DISH grade might be a reliable tool for evaluating the severity of this pathology. However, our grade may present challenges in the clinical setting. For example, there are exceptional cases in which bridging of OALL is found outside the thoracic spine. In addition, the clinical significance of this grading system is unclear. Thus, our future research will investigate the association between DISH grade and the incidence of vertebral body fractures.

The DISH grade is correlated with the cervical OPLL index and with the thoracic OLF and OSIL indices, all of which have frequently been detected in clinical settings [5,10,22,23]. Moreover, the progression of the DISH grade correlates moderately or slightly with various OSL indices, even in the whole spine. Thus, the severity of DISH might be correlated with that of OSL in other areas in the spine in patients with cervical OPLL. Okada et al. reported that surgery was performed in about 85% of cases exhibiting a spinal fracture with DISH, of which approximately 80% underwent conventional, open posterior fixation. In addition, the presence of OPLL was associated with residual neurological paralysis at the final follow-up [12]. In contrast, Yoshii et al. analyzed data from 2353 cases with cervical OPLL, of which 1333 cases underwent ADF and 1020 cases underwent PDF. Their report revealed that at least one local complication, such as cerebrospinal fluid leakage or surgical site infection, occurred in about 6.5% and 4.7% of anterior and posterior cases, respectively [24]. In cases with symptomatic cervical OPLL and/or DISH, including those with complications, the surgical outcomes were sometimes unsatisfactory; therefore, it is necessary to carefully monitor for neurological deterioration caused by the combination of multiple OSLs.

In this study, DISH was observed in nearly 40% of subjects and the most common grade was grade 2. This is because this study targeted patients with symptomatic cervical OPLL, and the range of DISH in the spine progressed with age. In contrast, fewer patients had grade 1. Although DISH is frequently comorbid with cervical OPLL [6,25], Fujimori et al. demonstrated that healthy subjects without OPLL may occasionally have DISH [5]. Therefore, our present findings and those of previous studies indicate that patients with grade 1 DISH can be broadly divided into two categories: those with and those without cervical OPLL. Our study focused specifically on patients with OPLL which could explain why a minority of the population had grade 1 DISH. Similarly, grade 3 DISH was an uncommon finding in our study population. As our results demonstrate, OSL may progress in patients with advanced DISH, and these patients usually need to be treated surgically. However, the present study did not include patients who had undergone spinal surgery, so the number of subjects with a grade 3 DISH was relatively small.

Ankylosing spondylitis (AS) is a spinal ankylosing condition similar to DISH. The radiological hallmark of DISH is ossification flowing along the spine similarly to “melting candle wax” [26,27,28], whereas AS is characterized by thinner and finer syndesmophytes connecting between adjacent vertebral bodies, which is known as “bamboo spine” [26,27,29]. Although experienced spine surgeons can easily distinguish between these two ossification disorders, it is uncertain whether all diagnoses are accurate. Moreover, these two pathologies occasionally show a degree of overlap [30]. Therefore, it is possible that our subjects diagnosed as having DISH constituted a heterogeneous population that consisted mainly of cases of DISH alone but may have also included some cases with AS or both conditions.

Spinal ossification is potentially associated with various metabolic diseases. In particular, DM is frequently comorbid with OSLs [31,32]; however, no significant correlation was found between the DISH grade and the prevalence rate of DM in the present study. A previous study found that the prevalence rate of DM was neither associated with the ossification types of OPLL nor the occupying ratio of OPLL in the spinal canal [33]. Thus, the presence or absence of DM might not be related to the radiographic progression of ossification.

This study has several limitations. First, our subjects were patients with symptomatic cervical OPLL who may have been predisposed to ossification in the whole spine. Further research is necessary to clarify whether our findings apply to asymptomatic patients with DISH found incidentally. Second, our study performed evaluations using CT imaging, which is associated with the problem of radiation exposure; therefore, it would be preferable to use plain radiography rather than CT. Finally, the study had a cross-sectional design, resulting in a lower level of evidence. Thus, a longitudinal study is needed to confirm whether the severity of OSL progresses with DISH simultaneously.

## 5. Conclusions

DISH was found in nearly 40% of patients with symptomatic cervical OPLL, about 60% of whom had a grade 2 DISH, using our classification system. Our DISH grade correlated with age and the indices of OSL in other areas at each high-prevalence level as well as the whole spine. Patients with cervical OPLL and severe DISH might also have a simultaneous severe OSL. In patients with symptomatic cervical OPLL, DISH extending to the cervical or lumbar spine is a radiographic sign suggesting a tendency toward diffuse ossification in the whole spine.

## Figures and Tables

**Figure 1 jcm-10-04690-f001:**
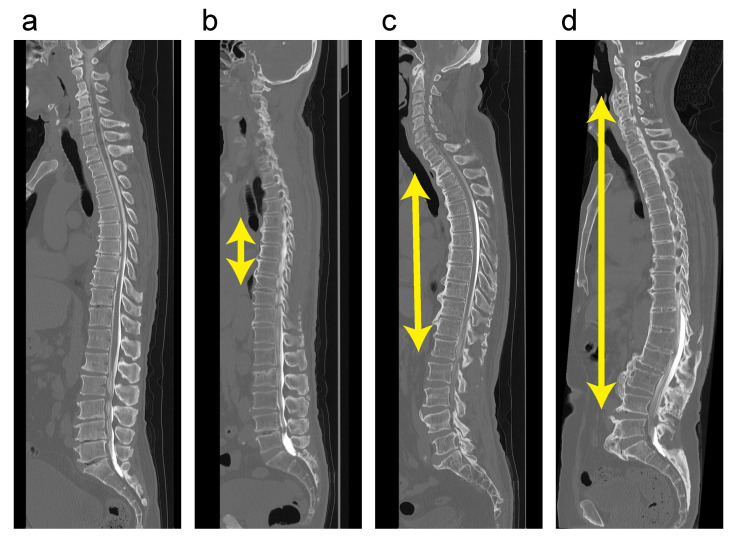
Representative sagittal computed tomography image for DISH grades 0–3. (**a**) Grade 0 (no DISH); (**b**) Grade 1 (DISH at T3–T10); (**c**) Grade 2 (DISH at both T3–T10 and C6–T2 and/or T11–L2); (**d**) Grade 3 (DISH extending beyond C5 and/or L3). DISH, diffuse idiopathic skeletal hyperostosis.

**Figure 2 jcm-10-04690-f002:**
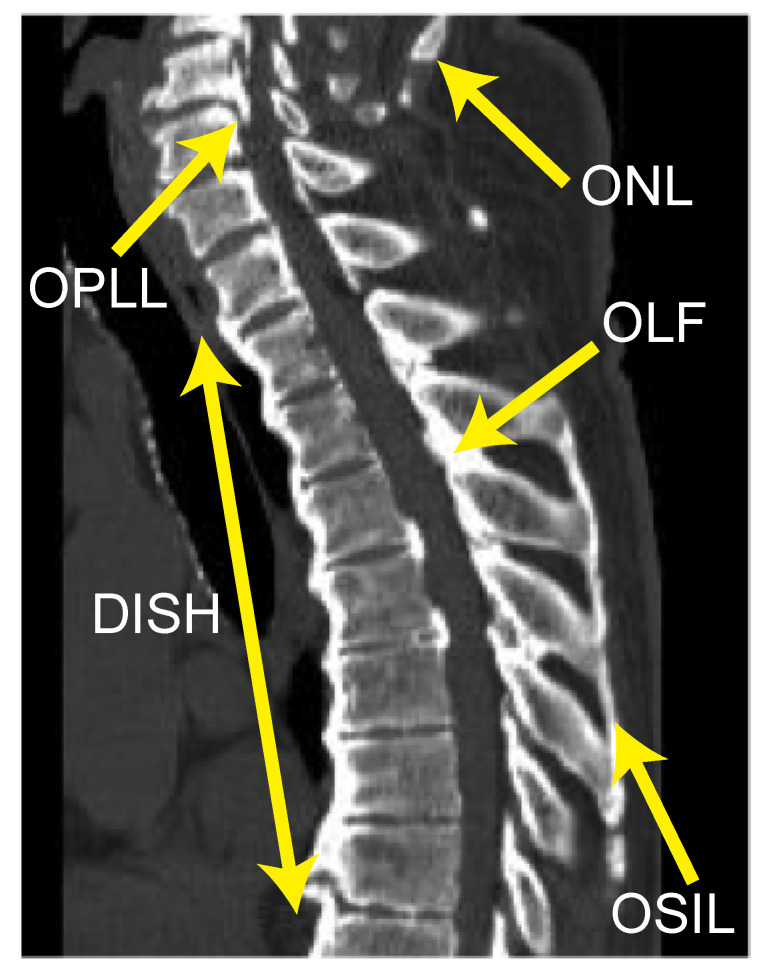
Representative sagittal computed tomography image for DISH, OPLL, OLF, OSIL and ONL. DISH, diffuse idiopathic skeletal hyperostosis; OLF, ossification of the ligamentum flavum; ONL, ossification of the nuchal ligament; OPLL, ossification of the posterior longitudinal ligament; and OSIL, ossification of the supra/interspinous ligaments.

**Figure 3 jcm-10-04690-f003:**
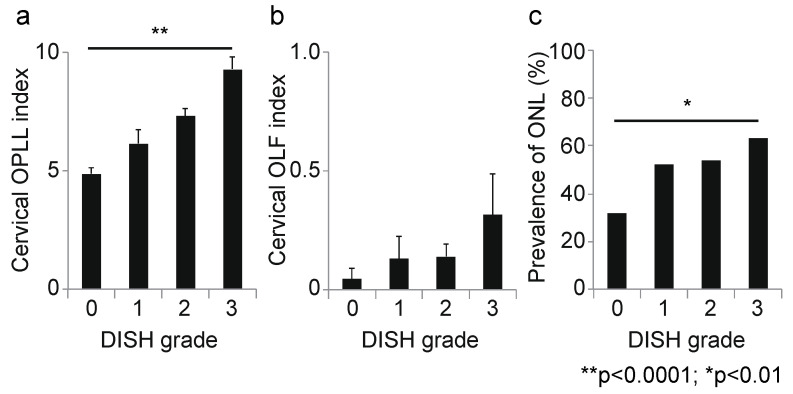
Correlation between DISH grade and OSL at the cervical spine. (**a**) Cervical OPLL index; (**b**) cervical OLF index; (**c**) prevalence of ONL. DISH, diffuse idiopathic skeletal hyperostosis; OLF, ossification of the ligamentum flavum; ONL, ossification of the nuchal ligament; OPLL, ossification of the posterior longitudinal ligament; OSL, ossification of the spinal ligaments.

**Figure 4 jcm-10-04690-f004:**
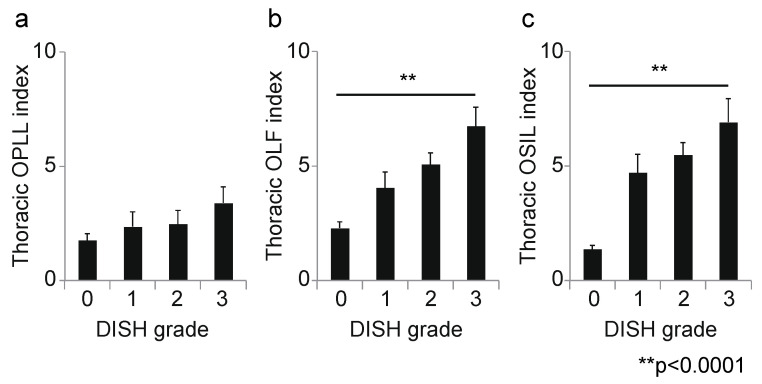
Correlation between DISH grade and OSL at the thoracic level. (**a**) Thoracic OPLL index; (**b**) thoracic OLF index; (**c**) thoracic OSIL index. DISH, diffuse idiopathic skeletal hyperostosis; OLF, ossification of the ligamentum flavum; OPLL, ossification of the posterior longitudinal ligament; OSIL, ossification of the supra/interspinous ligaments; OSL, ossification of the spinal ligaments.

**Figure 5 jcm-10-04690-f005:**
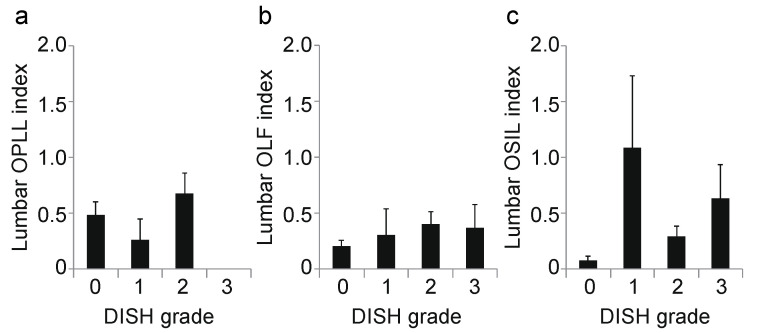
Correlation between DISH grade and OSL at the lumbar level. (**a**) Lumbar OPLL index; (**b**) lumbar OLF index; (**c**) lumbar OSIL index. DISH, diffuse idiopathic skeletal hyperostosis; OLF, ossification of the ligamentum flavum; OPLL, ossification of the posterior longitudinal ligament; OSIL, ossification of the supra/interspinous ligaments; OSL, ossification of the spinal ligaments.

**Figure 6 jcm-10-04690-f006:**
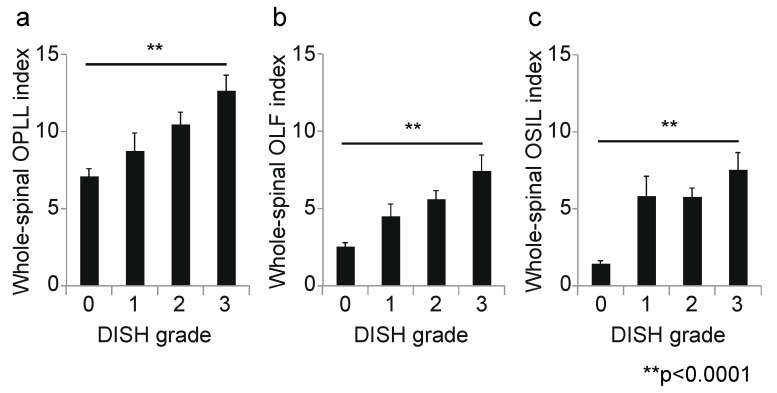
Correlation between DISH grade and OSL in the whole spine. (**a**) Whole-spine OPLL index; (**b**) whole-spine OLF index; (**c**) whole-spine OSIL index. DISH, diffuse idiopathic skeletal hyperostosis; OLF, ossification of the ligamentum flavum; OPLL, ossification of the posterior longitudinal ligament; OSIL, ossification of the supra/interspinous ligaments; OSL, ossification of the spinal ligaments.

**Figure 7 jcm-10-04690-f007:**
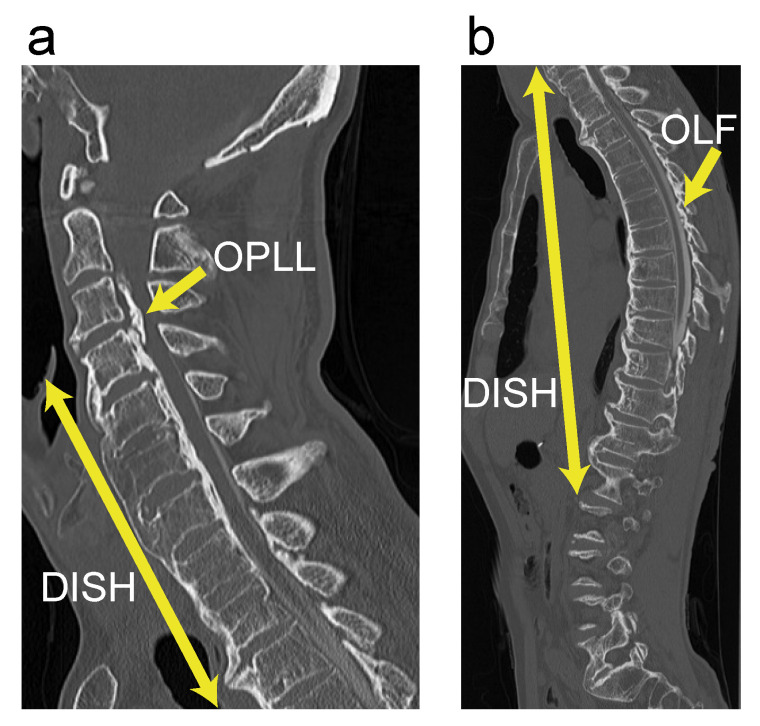
Illustrative case of grade 3 DISH. (**a**) Sagittal cervical CT imaging; (**b**) Sagittal thoracolumbar CT imaging. CT, computed tomography; DISH, diffuse idiopathic skeletal hyperostosis; OLF, ossification of the ligamentum flavum; OPLL, ossification of the posterior longitudinal ligament.

**Table 1 jcm-10-04690-t001:** Demographic data and surgery-related data for each DISH grade.

	Grade 0	Grade 1	Grade 2	Grade 3	*p*-Value
Patients, n	132	23	65	19	
Mean age (years)	60.9 ± 11.6	65.4 ± 12.7	66.9 ± 12.3	72.8 ± 9.9	<0.0001
Male sex (%)	61.4	78.3	75.4	78.9	0.09
Body mass index	26.1 ± 4.7	25.7 ± 5.0	25.9 ± 3.7	24.7 ± 4.9	0.32
DM (%)	21.2	30.4	32.3	15.8	0.25
FH of OPLL (%)	3.03	4.35	3.08	5.26	0.95
Trauma history (%)	6.82	0.00	7.69	10.5	0.53
Cervical level					
Patients treated surgically, n (%)	74 (56.1)	8 (34.5)	37 (56.9)	10 (52.6)	0.27
Surgical Method					
Laminoplasty (%)	40.5	62.5	51.4	60.0	0.40
Laminectomy (%)	1.35	0.00	0.00	0.00	0.86
ADF (%)	29.7	25.0	13.5	0.00	0.08
PDF (%)	27.0	12.5	32.4	30.0	0.71
APF (%)	1.35	0.00	2.70	10.0	0.37
Perioperative complication (%)	14.9	0.00	32.4	0.20	0.07
Neurological deterioration (%)	1.35	0.00	2.70	0.00	0.89
C5 palsy (%)	8.11	0.00	13.5	0.10	0.63
CSF leakage (%)	1.35	0.00	2.70	0.00	0.89
Surgical site infection (%)	0.00	0.00	5.41	0.00	0.17
Screw loosening (%)	1.35	0.00	0.00	0.00	0.86
Screw malposition (%)	1.35	0.00	0.00	0.00	0.86
Dysphasia (%)	0.00	0.00	2.70	0.10	0.10
Deep vein thrombosis (%)	0.00	0.00	2.70	0.00	0.47
Heart failure (%)	1.35	0.00	0.00	0.00	0.86
Delirium (%)	0.00	0.00	2.70	0.00	0.47
Thoracic level					
Patients treated surgically, n (%)	11 (8.33)	4 (17.4)	9 (13.8)	2 (10.5)	0.48
Surgical Method					
Laminectomy (%)	36.4	0.00	22.2	50.0	0.46
PDF (%)	54.5	75.0	66.7	50.0	0.86
PF (%)	9.09	25.0	11.1	0.00	0.79
Perioperative complication (%)	36.4	0.00	22.2	0.00	0.41
Neurological deterioration (%)	27.3	0.00	0.00	0.00	0.20
Surgical site infection (%)	9.09	0.00	11.1	0.00	0.88
Wound dehiscence (%)	0.00	0.00	11.1	0.00	0.58
Lumbar level					
Patients treated surgically, n (%)	6 (4.55)	2 (8.70)	3 (4.62)	0 (0.00)	0.62
Surgical Method					
Laminectomy (%)	33.3	50.0	66.7	0.00	0.63
PDF (%)	66.7	0.00	33.3	0.00	0.23
PF (%)	0.00	50.0	0.00	0.00	0.08
Perioperative complication (%)	0.00	0.00	0.00	0.00	-

ADF, anterior decompression with fusion; APF, anterior and posterior decompression with fusion; CSF, cerebrospinal fluid; DISH, diffuse idiopathic skeletal hyperostosis; DM, diabetes mellitus; FH, family history; OPLL, ossification of the posterior longitudinal ligament; PDF, posterior decompression with fusion; PF posterior fusion.

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
