# Peer review of "Association between Severity of Diffuse Idiopathic Skeletal Hyperostosis and Ossification of Other Spinal Ligaments in Patients with Ossification of the Posterior Longitudinal Ligament"

_jcm, 2021, doi:10.3390/jcm10204690_

Round 1
Reviewer 1 Report
Thank you for answering the Qs. I still personally doubt the scientific impact of this study. However, the authors successfully answered all questions. Therefore, I will leave the decision to the editor.
Reviewer 2 Report
Authors present prospective multicenter study of 239 patients with symptomatic OPLL and analyze DISH severity of CT of the total spine using the graduation system previously developed by the authors. DISH was found in 107 patients (44.8%), 65 (60.7%) of whom had 63 grade 2 DISH. Significant associations of DISH grade were found with the indices for cervical OPLL , thoracic ossification of the ligamentum flavum and thoracic ossification of the supra/interspinous ligaments.
This paper should be looked at as a rare possibility to hear the opinion of the true experts in the field, which is why additional changes need to be made. Authors raise a very important and so far not clearly understood question on ossification of the spinal ligaments, their clinical significance and correlation. It is especially important for the pathology of the cervical spine, as DISH can lead to symptomatic swallowing difficulties which require surgery and OPLL is known to be surgically treated using the dorsal approach - since the authors have more experience with this pathology then rest of the neurosurgical world, it would be interesting for the reader to include several illustrative cases and the modality of treatment (although this is not the primary intention of this paper). Furthermore, since symptomatic OPLL and symptomatic DISH usually require surgery, please include the data how many of the 239 patient were operatively treated, which technique was performed (open door laminoplasty, decompression, decompression + stabilization with lateral mass screws or ventral approach) and how many complications occured with , if possible, neurological outcome. If these data were already published in previous studies of the authors group, please cite these studies and shortly include the data in few sentences.
Since this is a prospective multicenter study, please include votum of the ethical comittee of the hospital(s) and/or health ministry for the study, including conduction of total spine CT for research purposes.
Please comment of one very important questions and include the data if applicable - in your opinion, is the ossification of the spinal ligaments result of a unknown metabolic or endocrinologic disease which is for some unknown reason localized in the spine, are there any papers on this subject and please include if possible a procentage of patients with OPLL/DISH who had metabolic diseases. Also please include procentage of patients with diabetes mellitus and procentage of smokers among the cohort, since there are several papers on the subject, try to find possible correlations and share your opinion on this not well understood entity.
Round 2
Reviewer 2 Report
The authors did not provide data on surgical treatment modality of the patients and on their smoking history; at least one of these two data should be provided.
Author Response
Please see the attachment.

This manuscript is a resubmission of an earlier submission. The following is a list of the peer review reports and author responses from that submission.
Round 1
Reviewer 1 Report
General comments
The authors have investigated the relationship between their custom-made DISH grading system and the incidence of various ossifications of the spinal ligaments (OSLs), including OPLL, OLF, and OSIL. They have found significant but rather weak to moderate, correlations between the DISH grades and age, cervical OPLL and thoracic OLF and OSIL as well as OSL indices in the whole spine. They have concluded that patients with a higher DISH grades should be investigated for OSL.
The biggest question that this reviewer has is the study subjects. The authors have employed patients with symptomatic cervical OPLL those who have not undergone surgery and this could be a source of selection bias. He is afraid to say that it may not be justified to claim that there is a significant correlation between DISH and cervical OPLL using this population. If the authors want to elucidate the true correlations between DISH and other OSLs, they should also include those with incidentally found asymptomatic DISH subjects.
The purpose of the study is unclear to this reviewer. Please explain how the development of this novel DISH grading system contributes to solve problems in our daily clinical practice.
Many indices and grades e.g., cervical OP index, OPLL index, cervical OP index grade, are used randomly, making reading and understanding of the context very difficult.
This reviewer suggests the authors to extensively revise the manuscript and resubmit.
Specific comments
The authors stated that "DISH is usually an asymptomatic incidental radiological finding," however, those with usual DISH are not included in the present study. (page 2, line 79)
What is the difference between "the number of consecutive vertebral bodies" and "the extent of ossification?" (page 2, line 87-8)
The authors vaguely stated that the aims of the study were to investigate the relationship between DISH grading and the frequency of OSLs, and to validate its usefulness in the clinical setting. Please explain what usefulness authors want to validate. How does this custom made grading affect our daily clinical practice? (page 3, line 100-2)
Were the assessments reproducible among six observers? (page 3, line 123-7)
The incidence of cervical OPLL must be 100% because patients with symptomatic OPLL were chosen as the study subjects. (page 3, line 124)
What do "the prevalence of ONL only was calculated for each grade?" and "OPLL, OLF, OSIL, and OALL for each vertebral body and intervertebral disc level" mean? What does "each" in these sentences mean? (page 3, line 127-9)
If the cervical OP index (ossification index of OPLL) were defined as the number of cervical levels with OPLL, why this index exceeds 8? The index must be seven at maximum. Or did the authors added vertebra and disc levels? If so, this must be clearly explained in the text. (page 4, line 131-3)
"OSL index" suddenly appears here without definition. (page 4, line 146)
Correlation coefficient of 0.3 indicates a weak correlation only and may be less clinically relevant. (page 5, line 156)
There are too many indices/grade; e.g.; cervical OP index, cervical OP index grade, OPLL index, and OSL index, which are confusing and makes understanding very difficult. (page 5, line 164-6)
Correlation coefficients >0.7 indicate strong, 0.7-0.4 moderate, and 0.4> weak. Most coefficients here represent moderate to weak correlations, especially 0.29, although they may be statistically significant. (page 5. Line 164-74)
The authors found a significant but weak correlation (r=0.3) between DISH grade and age. Please explain what this finding mean in our daily clinical practice. (page 7, line 203-5}
Again, the correlation coefficients between DISH grade and OSLs, especially OPLL (r=0.29) are only marginal. (page 7, line 208-9)
Are the discussions on Wnt inhibitors, osteocalcin and Dickkopf-related protein, relevant to the results of the present study? (page 7, line 211-6)
The authors stated that ossification elsewhere in the spine should be suspected in a patient with a progressing DISH grade. However, the base population of this study is those with symptomatic cervical OPLL and there is a possible selection bias. This reviewer is not sure if this comments can be deduced from the results of the present study. If the authors want to stress this point, they should include subjects with incidentally found DISH and reassess the correlations with different OSLs. (page 7, line 218-9)
This reviewer is further confused by the authors' comment "The clinical significance of this grading system is unclear." What are the practical and substantial aims of the study, and therefore, clinical relevance of the grading system? (page 8, line 250)
Please explain how this grading system solves the problem of how to assess the progression of DISH with aging, and how this affects our clinical judging when seeing patients with OSLs. (page 8, line 257-62)
Chicken or the egg logic? To find patients with a high DISH grade, a whole spine CT is necessary anyway. (page 8, line 260-1)
Reviewer 2 Report
Authors investigated the rate of various types of ligament ossification.
- First of all, I suspect that all DISHs start from thoracic spine. There are patients who only have cervical DISH. As such, I do not think the methodology is right.
- The inclusion criteria are biased. Authors chose symptomatic patients with cervical OPLL but underwent no surgery. This group has a certain trend, not reflecting a general population. External validity would be poor.
- It is obvious that patients with OPLL tend to have DISH and/or other ligament ossification and vice versa. I suspect the impact of this study.
- What is the rationale of choosing C5/L3 as the border level?
- What if a patient had DISH only cervical spine?
Overall, the manuscript is not logical and lacks scientific importance.
Round 2
Reviewer 2 Report
Thank you for answering all my questions. The paper is now better than the previous one.